# Use of Different Types of Magnetic Field Sensors in Diagnosing the State of Ferromagnetic Elements Based on Residual Magnetic Field Measurements

**DOI:** 10.3390/s23146365

**Published:** 2023-07-13

**Authors:** Maciej Roskosz, Paweł Mazurek, Jerzy Kwaśniewski, Jianbo Wu

**Affiliations:** 1Department of Machinery Engineering and Transport, Faculty of Mechanical Engineering and Robotics, AGH University of Science and Technology, 30-059 Krakow, Poland; pmazurek@agh.edu.pl (P.M.); kwasniew@agh.edu.pl (J.K.); 2Department of Engineering Science and Mechanics, Sichuan University, Chengdu 610065, China; wujianbo@scu.edu.cn

**Keywords:** non-destructive testing, magnetic sensors, residual magnetic field, steel wire rope

## Abstract

The early identification of micro-defects in ferromagnetic elements such as steel wire ropes significantly impacts structures’ in-service reliability and safety. This work investigated the possibility of detecting mechanically introduced discontinuities using different magnetic sensors without magnetization of the tested object with a strong external field. This is called the passive magnetic testing method, and it is becoming increasingly popular. This research used differential sensors (measuring differences in field values at the nanotesla level) and absolute sensors (enabling the measurement of the magnetic field vector module or its components at the microtesla level). Each measurement result obtained from the sensors allowed for detecting discontinuities in the line. The problem to be solved is the quantitative identification of changes in the metallic cross-section of a rope.

## 1. Introduction

### 1.1. The Current State of Steel Wire Rope Diagnostics

The first steel wire rope in the world was made by Wilhelm Albert in the 1800s. Steel wire ropes are used in every industry (personal lifts, cableways, mining). The most critical defects are wire breakage, strand breakage, rust, wear and fatigue [1]. Therefore, much work is connected to the investigation of various wire rope failure detection methods to guarantee their reliability and safety [2]. Many researchers have raised the problem of magnetic wire rope diagnostics. It is a fundamental problem, and it is not easy to unequivocally systematize. An up-to-date and extensive overview of non-destructive testing methods used to diagnose steel ropes was presented in [3]. Magnetic flux leakage is the most useful non-destructive testing method currently known and used for wire rope testing. Nevertheless, there are other well-known methods; these include eddy current testing, ultrasonic guided waves, acoustic emission, radiography and vision testing. Each method uses different physical phenomena; intensive research is conducted on almost every method.

Increasing interest has been shown in the development of new methods for diagnosing steel wire ropes. In [4], the authors designed a device prototype based on the measurement of residual magnetic field (RMF) components. The authors proved that the proposed model is better than traditional methods of detecting wire rope discontinuities due to its high accuracy and the small dimensions of the sensor. The paper presented an algorithm based on the Hilbert–Huang transform (HHT) and compressed sensing (CS). The experimental results showed a very high convergence of the damage indicated by filtration with the actual damage. However, these results are insufficient because the authors ignored the direction of the damage on the circumference of the rope. In engineering practice, the direction of damage is vital because damage located along the entire rim of the rope is less dangerous during operation than damage occurring along the same direction. The paper did not consider the rope pitch, a spiral resulting from twisting the strands making up the rope. The research in [5] used a similar wire rope discontinuity detection principle, wherein the researchers placed 18 GMR sensors around the perimeter of the rope. It made a significant contribution to the development of the diagnostic method. Additionally, permanent magnets were used in the work to strengthen the diagnostic signals, and, despite its complexity, the applied wavelet filter ensemble empirical mode decomposition (EEMD) effectively reduced the noise. Unfortunately, the work did not show a correlation between the signal and the damage, i.e., we cannot say anything about the size and shape of the damage. The authors attempted to diagnose wire ropes on real objects in real time in subsequent works. For diagnostic and operational reasons, it is essential to diagnose rope discontinuities (damage) and the stresses that occur. Ref. [6] described the use of a sensor with the function of vibration dissipation. It is crucial to diagnose working objects as vibrations effectively hinder correct diagnostics. Unfortunately, the obtained results were not satisfactory. The authors of [7] focused on the quantitative diagnosis of discontinuities. Instead of the traditional Hall sensor system, a magnetic concentrator was used. This sensor allowed for comprehensive data collection. Experimental studies were carried out on discontinuities in one to five wires. The analysis of the results was inconclusive, and additional studies are required to implement the proposed method successfully. Ref. [8] proposed a convolutional denoising autoencoder (CDAE) and isolation forest (iForest). The work in that study is innovative due to the device learning algorithm for the occurring defects, and the proposed method can catch physical, structural and material defects. Unfortunately, this method is limited to training, with a few common drawbacks. In practice, each failure is different, so the proposed algorithm may not be able to function effectively. The research in [9] considered a rope’s pitch, i.e., the twist angle resulting from its production. The focus of that paper was on the surface defects of the rope, which are significant in closed ropes. Thanks to the use of complex algorithms and the division of the rope sections into segments, high efficiency was achieved. An innovative research technique allowed for the detection of defects in challenging environmental conditions, such as grease, dust and dirt. Ref. [10] used magnetic flux leakage (MFL) and introduced the slotted ferromagnetic lift-off layer. Compared to the conventional air gap method, the ferromagnetic gap has proven more effective, with the gap in the ferromagnetic layer strengthening the diagnostic signal. However, there is a problem with diagnosing damaged components or those with a complex structure because the introduced ferromagnetic layer could also be damaged during the measurement. Ref. [11] considered not only the detection of defects distributed along the circumference of wire ropes but also the angle of inclination of wire ropes concerning the sensor. Unfortunately, the traditional Hall effect sensors in typical MFL detection arrangements used here have large dimensions, which makes it very difficult to measure the conditions of the real object. Ref. [12] compared the alternating current magnetic flux measurement (AC-MFM) method with the permanent magnet magnetic flux measurement (PM-MFM) method. The PM-MFM method is superior to AC-MFM, mainly due to the speed of its measurements. The experimental analysis confirmed that the proposed PM-MFM sensor is compatible with many ferromagnetic materials, which allows it to be widely used in industry. The authors of [13] used different eddy-current-based probe designs—total and commercial reflection—to detect defects. Their experimental results showed that the total probe is suitable for detecting cracks and holes. In contrast, the reflection probe is more suitable for detecting subsurface defects, such as small-diameter blind holes. This is a critical observation when detecting defects on the surface and inside the tested object. In [14], theoretical and experimental studies were conducted to determine fatigue life as a criterion for depositing a non-rotating rope through alternating bending. As a result of empirical studies, it was proven that the rope’s service life decreases with the load and the reduction in the diameter of the pulley through which it is bent. A very high correlation between the experimental results, artificial neural network (ANN) and the results of the proposed regression model was found. Ref. [15] proposed an innovative approach to diagnosing wire ropes based on texture characteristics. Attention was paid to the selection of lighting and appropriate filtering algorithms. Unfortunately, these algorithms only focus on the defects in the outer strands of the ropes and do not consider the imperfections in the wires inside the rope. Diagnostics of steel ropes used as load-bearing elements of bridges were discussed in detail in [16]. Magnetic flux detection was used for the measurements. The model was analyzed using the finite element method. Here, high efficiency in detecting damage was demonstrated, but the damage corresponded to defects in the wires present in the outer layers of the rope. In addition, the magnetic head used in that study has large dimensions, making it difficult to perform measurements in real conditions on site. The authors of [17] paid particular attention to the stage of appropriate analysis of the obtained diagnostic signal of a wire rope. Eight noise reduction and signal processing methods were proposed: low-pass, Butterworth, median and mean filtering, Gaussian, polynomial fitting, wavelet and empirical mode decomposition (EMD). The Gaussian filtering method showed the best performance. Unfortunately, too few wire rope constructions were explored, preventing translation into a universal algorithm. The study’s authors emphasized the possibility of combining various signal analysis methods, which requires further work and research. The authors of [18] presented a detailed analysis of magnetic field reconstruction using a pulsed wire method. This is an innovative method that depends on the length of the pulses. Despite the numerical simulations, the results obtained were unreliable. This method needs to be refined. The authors of [19] drew attention to the shortcomings of the proposed innovative pulsed wire magnetic method. Due to the unique structure of the frame, measurements on real objects are impractical and, at the same time, expensive. This method is also burdened with internal sources of errors that distort the obtained results’ credibility. Despite the signal analysis algorithm proposed by the authors, it still requires a very detailed analysis to be used in practice.

### 1.2. Aim of this Work

Thus far, none of the existing studies have compared passive magnetic sensors in the context of wire rope diagnostics. This paper aimed to analyze the diagnostic capabilities of sensors using the following phenomena: magnetoimpedance (MI), tunneling magnetoresistance (TMR) and optically pumped magnetometers (OPMs). In this work, particular emphasis was placed on the influence of the distance between the sensor and the tested object, and the direction of the test. The measurement results obtained from various sensors were compared qualitatively and quantitatively. This article aimed to indicate the most appropriate sensor that best shows the modeled failure description as a beginning for further diagnostic calculation algorithms. The traditional magnetic method of wire rope testing is based on formatting the magnetic field from an external source and analyzing the magnetic flux leakage caused by discontinuities [4,20,21,22]. The techniques we propose use residual magnetic fields (self-magnetic flux leakage), which result from the mechanical and structural inhomogeneities of the material [23,24,25].

## 2. Materials and Methods

### 2.1. Examined Object—Steel Wire Rope

The correlations described in this article were found for many structures of the tested wire rope. This work focused on a test for a representative example of the tested object described here. The tested object was a steel rope with a diameter of 6 mm and a 6 × 19 S + IWRC (7 × 7) construction. This structure is often used in industry due to its flexibility, especially in passenger lifts. The metallic cross-section of the tested object and the strength characteristics are shown in Figure 1.

The rope has a 6 × 19 S + IWRC (7 × 7) construction, which means that the rope consists of 6 strands of 19 wires each. The strands have a Seale construction. Thomas Seale patented this design in 1855 in the USA. The Seale strand is characterized by a layer of wires with smaller diameters laid directly on the core, with the next layer consisting of the same number of wires with the same but much more prominent (close to the core) diameters [26]. It causes an increase in fatigue resistance to bending and elasticity. These features are essential in industrial applications—mainly in the lifting industry. The independent wire rope core (IWRC) is also made of seven strands, with seven wires each. Using a steel rope as the core increases the tensile strength compared to a rope with a fiber core.

### 2.2. Self-Magnetic Flux Leakage Method

The subject of ongoing research is the analysis of the possibility of using the passive magnetic technique in diagnosing ferromagnetic objects [27]. Ref. [28] proved that the residual magnetic field is an effective carrier of information about the technical condition of suspension ropes. The self-magnetic flux leakage (SMFL) method is a passive variant of the magnetic flux leakage (MFL) method [4]. The MFL method is described in detail in [29]. The main difference is that the SMFL method relies on the self-magnetization of ferromagnetic material in a geomagnetic field, while the MFL method requires an externally formatted excitation source [30]. In [31], three theoretical models were proposed to explain flux changes: the concentration domain model (CDM), the rotation domain model (RDM) and the vertical domain model (VDM). Finally, the authors concluded that the theoretical models and experimental results showed very similar trends.

The SMFL technology creator is AA Dubov, who described this phenomenon as metal magnetic memory (MMM) [32,33]. He was the first to notice the relationship between a ferromagnetic object’s magnetic and mechanical properties under different loads [34]. The physical essence of the MMM technique is the magnetic/stress coupling effect, also called the magneto-mechanical effect [35]. The MMM technique combines a geomagnetic field and mechanical stress due to defects and stress, utilizing the spontaneous magnetization induced by ferromagnetic materials. The magnetic flux is not disrupted if the object is free of abnormalities [36]. The results presented in [37] confirmed the possibility of using an algorithm in steel elements regardless of the orientation and depth. The interpretation of the signal requires carrying out further research, especially numerical modeling. In [38], ANSYS was used to simulate a wire rope model, and the magnetic memory signal change characteristics of the wire rope with early damage under a weak magnetic field and geomagnetic field were studied. The authors concluded that combining the normal component’s peaks and valleys with its gradient value makes it possible to judge the stress concentration and early damage of wire ropes. Although researchers have given various explanations for the magnetic force coupling mechanism, there still needs to be a systematic theoretical explanation for the mechanisms of the elastic and plastic stages. In [39], a dual-defect magnetic dipole model was proposed to describe wire breakage in bridge cables. SMFL signal detection experiments were carried out with axial, radial and circumferential wire breakage gaps. The authors of [40] emphasized that a significant threat to this method is the influence of the magnetic force of the neighboring elements of the tested object, which can effectively suppress the diagnostic signal. The researchers of [41] rightly noted that, in the case of passive crack monitoring, it is essential to study and quantify the effect of large and cyclic stresses in weak magnetic fields. Magnetic signal inspection based on the self-magnetic flux leakage (SMFL) effect can effectively identify the location of defects. However, current research on the magnetic signal of defects under the influence of various factors needs to be more comprehensive. Based on the SMFL inspection of defective steel wires under tensile load, ref. [40] proposed an approach to assess defects using the peak value of the magnetic signal’s normal component Hp(y). These assumptions were confirmed in [42], where the depth of damage was dependent on the amplitude of the diagnostic signal. A detailed signal analysis with promising conclusions was presented in [43]. Many studies have been conducted on corrosion detection in load-bearing structures [44,45]. In [39], a dual-defect magnetic dipole model was proposed to describe wire breakage in bridge cables. SMFL signal detection experiments with various radial, axial and circumferential wire crack width measurements were carried out. As shown above, this method is the subject of intense research. However, no papers that compare different techniques and sensors used to measure the magnetic signal can be found.

## 3. Sensors

### 3.1. Types of Used Sensors 

The basis of this article is the variety of passive magnetic sensors. This diversity results from other physical phenomena used for the described sensors. This work compared all magnetic signatures from magnetometers available in the laboratory, designated as S1, S2, S3 and S4. The characteristics of these sensors are shown in Table 1.

The S1 and S2 sensors can detect magnetic field variation at the nanotesla level. The sensors consist of a 1-axial magnetic head (MI element) and an electric circuit to operate the MI element. By restricting the cut-off frequency on the low-frequency side to 0.1 Hz, the sensors cancel static magnetic fields such as geomagnetism and only respond to moving ferrous objects with high sensitivity. 

The S3 sensor is a micro-fabricated atomic magnetometer (MFAM). A miniaturized scalar atomic magnetometer (sensing module) was built to measure changes in the Earth’s magnetic field associated with natural and artificial phenomena. The MFAM sensing module includes two laser-pumped cesium sensors connected to the sensor driver electronics. 

The S4 sensor is a low-power, 3-axis USB digital vector triaxial magnetometer based on the magnetoresistance tunneling effect (TMR) and quantum magnetic tunnel junctions (MTJs). 

### 3.2. Calibration and Validation of Used Sensors 

Sensors S1, S2 and S3, due to their design and measurement methods, do not require calibration, and even their calibration is impossible. In contrast, the S4 sensor requires calibration. Quantitative results of measurements carried out with this sensor depend on its calibration, which can be carried out for any magnetic condition. The calibration process causes the sensor to adopt local values at the calibration site of absolute 0.

The essential device involved in the S4 sensor calibration process was the Zero Gauss Chamber 12″ ZG-212, made of a mu-metal alloy (an alloy of 75% nickel and 15% iron), from the Magnetic Shield Corporation (USA). Thanks to the three-layer construction of the chamber, the external magnetic field and the Earth’s geomagnetic field (0.25–0.65 Gs) are suppressed to the level of mGs.

The S4 sensor was calibrated in the ZG-212 chamber, treating the value of the magnetic field in it as zero.

To verify the indications of the S4 sensor, a Helmholtz coil with laboratory amplifiers was used to generate the appropriate current intensity values. The relationship between the coil’s current and its magnetic field is shown in Figure 2.

The S4 sensor was placed in the center of the coil, and measurements were carried out on the changing magnetic field generated by the windings of individual coils. The influence of ambient conditions (the values of magnetic induction components at the measurement site resulting from the impact of the Earth’s magnetic field and the immediate environment) was considered. The measurement results of the field values are shown in Figure 3, and the values of the absolute and relative errors are shown in Figure 4 and Figure 5, respectively.

## 4. Experimental Details

Each of the sensors, i.e., S1, S2, S3 and S4, was used simultaneously for the magnetic field measurement (Figure 6). Sensor S4 was calibrated only once at the beginning of the rope at a distance of 25 mm (series 1—Figure 7). Each magnetic sensor was placed on a test strand (Figure 6). The rope was cut into pieces 700 mm long. Loops were formed at the ends to be attached to the test stand (Figure 7). The measuring range of the rope was 150 mm. The rope realized a movement with a constant speed of approx. 3 mm/s. The measuring range of the steel wire rope is shown in yellow. The laboratory where the examination was performed did not contain other ferromagnetic elements that could affect the result. Three series of measurements were carried out, with five measurements in each series. The distribution of the components of magnetic induction along the rope was measured and recorded in both movement directions of the rope (a—forward direction; b—backward direction). The average (mean) of all values was used for the final analysis. 

In series 1 of the measurement, the steel wire rope was installed in its delivered state on the stand, according to Figure 7. The distance between the sensors and the rope was 25 mm in the z-direction and 0 mm in the x- and y-directions (Figure 7). The distance between sensors S3_1 and S3_2 was 10 mm. 

Series 2 consisted of measurements of the rope in which the test object was changed by introducing, in the middle of the rope, a discontinuity to a depth of approximately 2 mm (Figure 8 and Figure 9). The localization of the sensors relative to the tested object stayed the same as that of series 1. 

The difference between series 3 and series 2 is the change in the lift-off value. The sensors were brought closer to the rope to a distance of 10 mm. At the same time, the S3_1 and S3_2 sensors were replaced, and the distance between them was set to 50 mm (Figure 10). 

## 5. Results

The results of the conducted measurements are shown in Figure 11, Figure 12, Figure 13, Figure 14, Figure 15, Figure 16 and Figure 17. The designations used in the drawings consist of the series number and the direction. In these figures, marking _1a is assigned to the first series, in the “forward” direction; marking _2a is assigned to the second series, in the “forward” direction; and marking _3a is assigned to the third series, in the “forward” direction. Similarly, markings _1b, _2b and _3b are assigned to the respective series of measurements (1, 2 and 3), but in the opposite “backward” direction.

## 6. Discussion

Although we measured the same range of the rope under the same conditions, the results were different, which is understandable due to the measurement techniques of each sensor.

S1 sensor (Figure 11): Compared to the other sensors, the obtained magnetic induction difference values were minimal—several dozen times lower. The change in the magnetic signal was not recorded at a distance of 25 mm from the sensor—only for a distance of 10 mm. The sensor registered changes in induction in the vicinity of the discontinuity, but these changes were still present in a large area behind the discontinuity. Near the discontinuity, the amplitude of the signal changes was the largest, which allowed for its localization. The recorded changes did not correspond to the natural changes in the induction value (which can be determined based on the measurement results of the S4 sensor). It is very puzzling that the signals differed in opposite directions, i.e., there were different amplitudes. There were additional disturbances in the signal waveform, the reasons for which we cannot determine; however, one of the reasons is probably the vibration of the sensor.S2 sensor (Figure 12): We can see an extensive analogy here regarding the comments concerning the S1 sensor, which uses the same measurement method. It is evident here, however, that the signal’s amplitude was lower at this distance from the test object, probably due to the different designs and configurations of the measuring coils. Qualitatively, the course of the recorded signal distribution was very similar to the distribution of the signal from the S1 sensor, but quantitatively, this signal was about twice as small. As in sensor S1, the discontinuity caused changes in the signal, and all the comments made for sensor S1 apply here.S3 sensor (Figure 13 and Figure 14): In the case of the S3_1 and S3_2 sensors, we observed the high convergence and repeatability of the measurement. The S3 sensor, in contrast to the S1 and S2 sensors, measures the absolute value of the magnetic induction vector read at a specific point on Earth. The results were several dozen times higher than those of the S1 and S2 sensors. However, the sensor’s sensitivity deserves attention—the damage was detected at 25 mm from the rope (sensors S1 and S2 did not indicate damage at this distance). An apparent signal change could be seen concerning the signal for the rope with damage. Compared to the distribution of induction measured for the rope without discontinuities, the introduced discontinuity caused a change in the distribution in the form of a local anomaly of a sinusoidal function. The location of the discontinuity coincided with the local minimum of this anomaly. Unfortunately, this sensor also has its limitations. For the distance between sensors S3_1 and S3_2 equal to 10 mm (as in series 1 and 2) and the distance of sensors from the rope also equal to 10 mm (third series of measurements), the measurement range was exceeded, which made it impossible to obtain a physically possible induction distribution for this case. The sensors had to be moved horizontally to a distance of 50 mm, as shown in Figure 10 (series 3). In most measurement cases, the induction distributions measured by the S3_1 and S3_2 sensors were similar in quality, with quantitative differences. In one case, the results for sensor S3_2 differed both qualitatively and quantitatively.S4 sensor (Figure 15, Figure 16 and Figure 17): The S4 sensor seems to be the most promising in detecting discontinuities in wire ropes. It is worth noting that changes resulting from discontinuities in the steel rope were visible at 10 mm and 25 mm distances. Moreover, these changes were visible in every component of the signal. It is also worth noting that the direction of the measurement had no effect on the obtained result, which was the same for the two opposite directions. The magnetic anomaly caused by discontinuities for each component caused a characteristic distribution disorder. In the location of a discontinuity, tangential components (parallel and perpendicular to the axis of the rope—references) have local extremes. In the distribution of the normal component, there are two local extremes (maximum and minimum), and the distribution of values in its vicinity has a sinusoidal shape. The distributions of the tangent components parallel to the rope axis and the normal (radial) tangent provide the most visible diagnostic information.

## 7. Conclusions

This research aimed to determine the possibility of using magnetic gauges to detect discontinuities in steel ropes. Three series of measurements were carried out, including measurements on ropes with and without discontinuities, using various configurations of the arrangement of the measuring sensors used concerning the test object. It can be concluded that the results obtained from each tested sensor can be used as diagnostic signals in steel rope defectoscopy. The different magnetic field measurement methods used in these sensors and differences in their design mean that the received diagnostic signals—their distribution along the measuring section of the rope—differ significantly in terms of quality and quantity. The critical issue concerning the diagnostics of wire ropes is the ability to identify signal changes that differ for individual sensors due to discontinuities. These changes result mainly from the measurement method used in a given sensor. Despite these differences, the measurement results of each sensor allow an approximate determination of the location of a single discontinuity.

In the case of sensors S1 and S2, the location of the discontinuity was clearly distinguished in the signal waveform, which showed a value close to 0 outside this location. The signal-to-noise ratio for sensors S1 and S2 was high and very satisfactory regarding their use in wire rope flaw detection.

The S3 and S4 sensors, in contrast to the S1 and S2 sensors, measure the value of the magnetic induction vector read at a specific point on Earth. A discontinuity introduced on the rope’s surface caused a local magnetic anomaly in the induction distribution. This was particularly visible in the image of the induction components obtained from measurements with the S4 sensor.

The sum of the diagnostic information obtained from the tests allows us to conclude that the mutual complementation of sensors using different measurement methods allows for the development of diagnostic tools that can be used in the passive diagnostics of steel ropes. Separate issues and subjects of further research are issues related to the configuration of the sensor or sensors relative to the rope (lift-off), determining the size of the discontinuity based on the diagnostic signal, the impact of external factors and the resolution in identifying closely located defects.

## Figures and Tables

**Figure 1 sensors-23-06365-f001:**
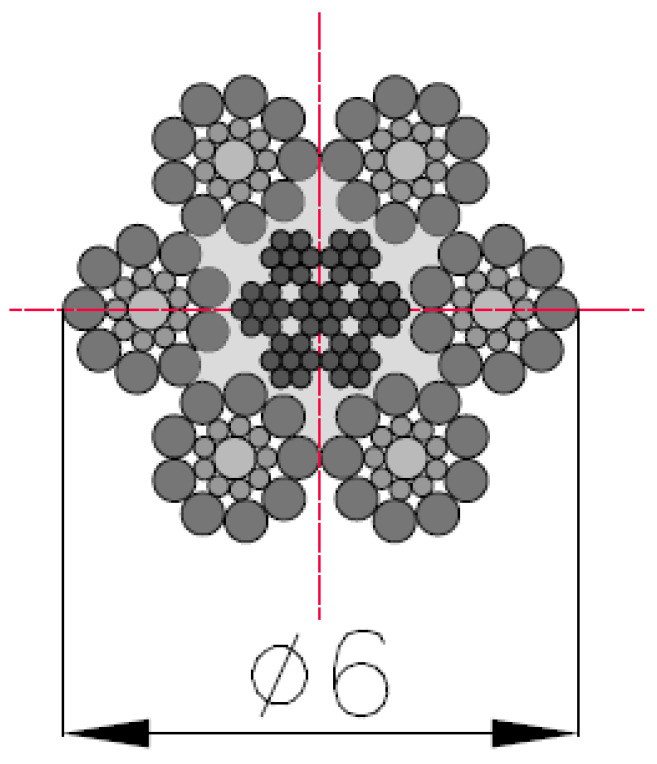
Cross-section and details of the measured steel wire rope.

**Figure 2 sensors-23-06365-f002:**
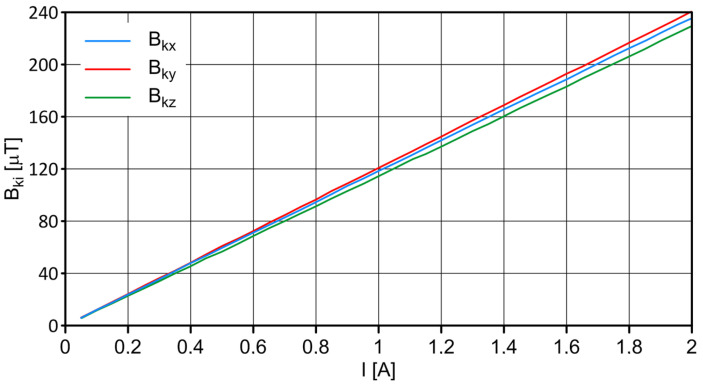
Helmholtz coil calibration curves.

**Figure 3 sensors-23-06365-f003:**
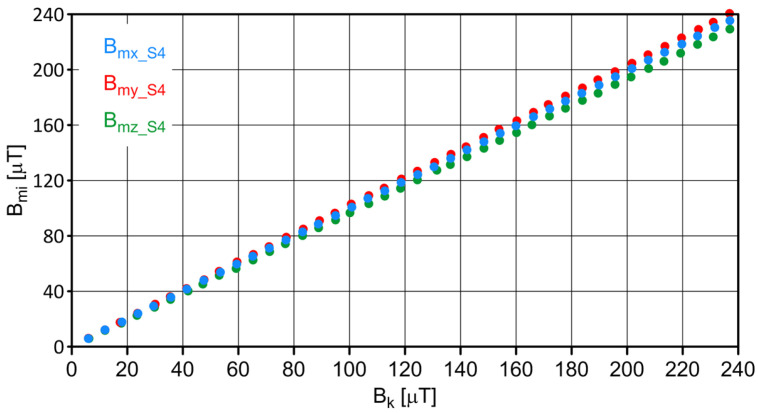
Measured B_mi_ values for sensor S4.

**Figure 4 sensors-23-06365-f004:**
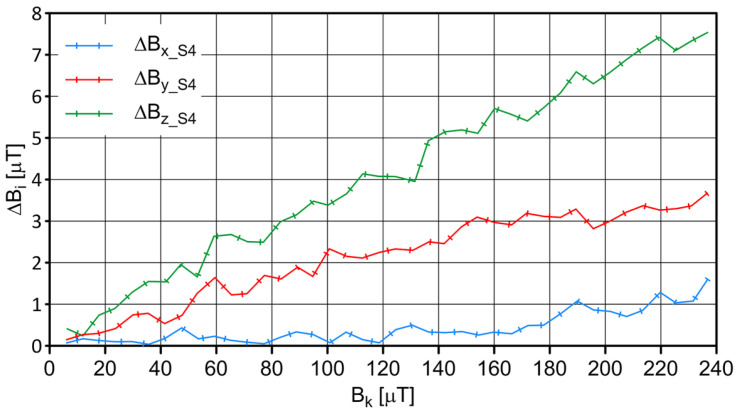
The absolute error of indications, which depends on the magnetic field’s value for sensor S4.

**Figure 5 sensors-23-06365-f005:**
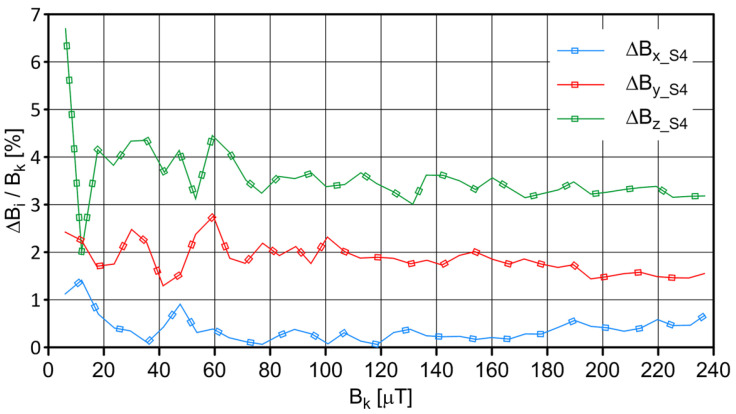
The relative error of indications, which depends on the magnetic field’s value for sensor S4.

**Figure 6 sensors-23-06365-f006:**
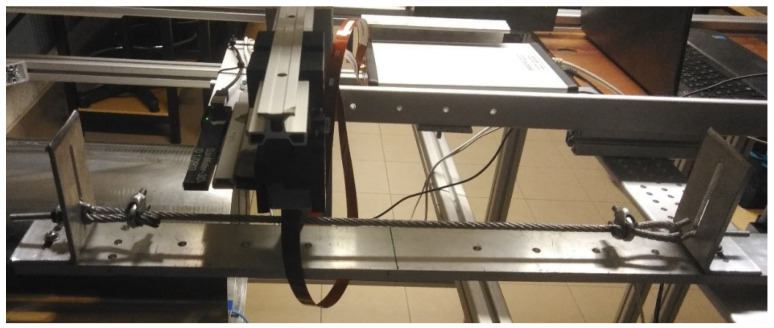
Real test stand.

**Figure 7 sensors-23-06365-f007:**
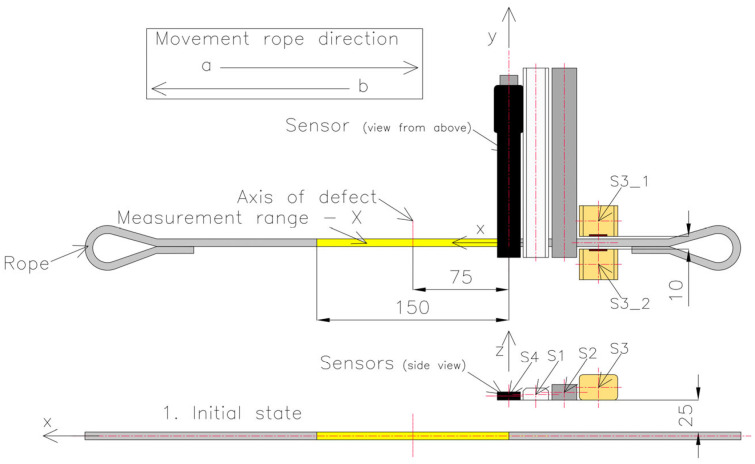
Setup of the measurement system in series 1.

**Figure 8 sensors-23-06365-f008:**
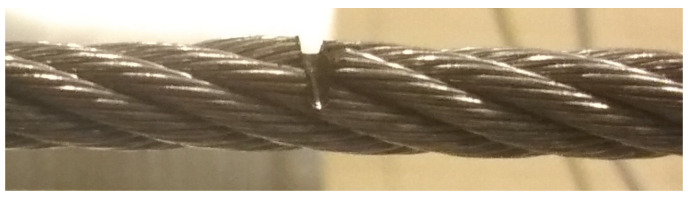
Real view of the artificially introduced discontinuity in the rope.

**Figure 9 sensors-23-06365-f009:**
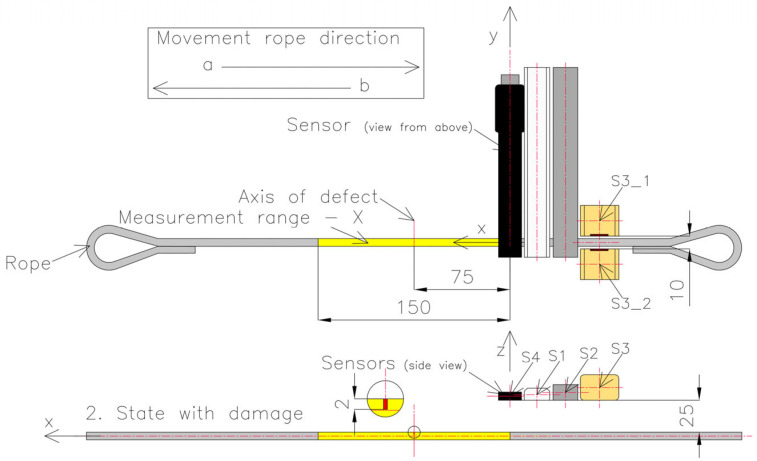
Setup of the measurement system in series 2.

**Figure 10 sensors-23-06365-f010:**
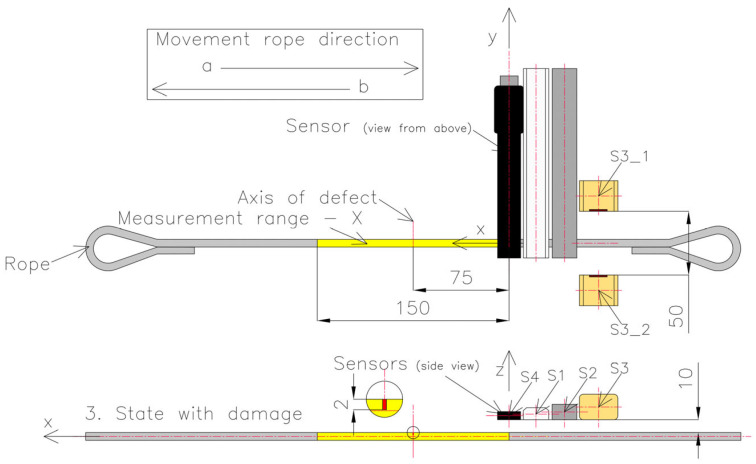
Setup of the measurement system in series 3.

**Figure 11 sensors-23-06365-f011:**
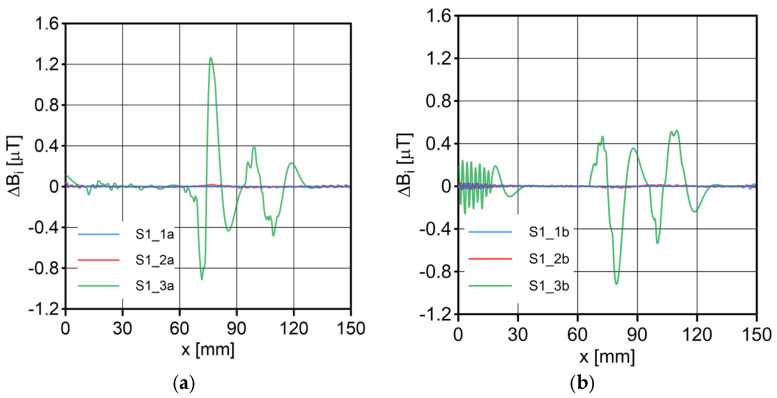
Distribution of induction difference ∆*B* measured by sensor S1 along the rope: (**a**) forward direction; (**b**) backward direction.

**Figure 12 sensors-23-06365-f012:**
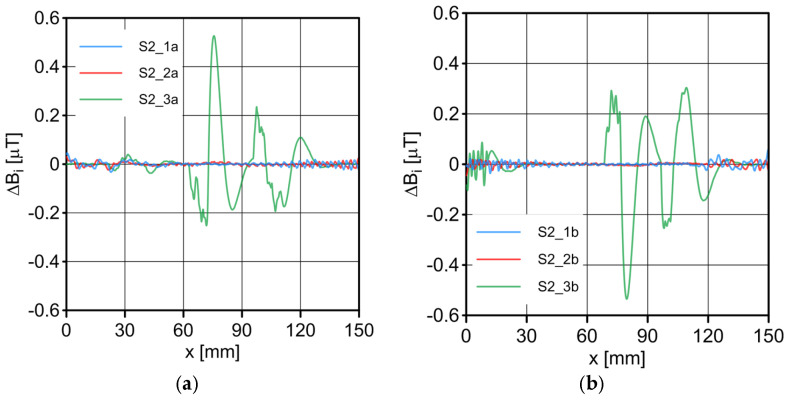
Distribution of induction difference ∆*B* measured by sensor S2 along the rope: (**a**) forward direction; (**b**) backward direction.

**Figure 13 sensors-23-06365-f013:**
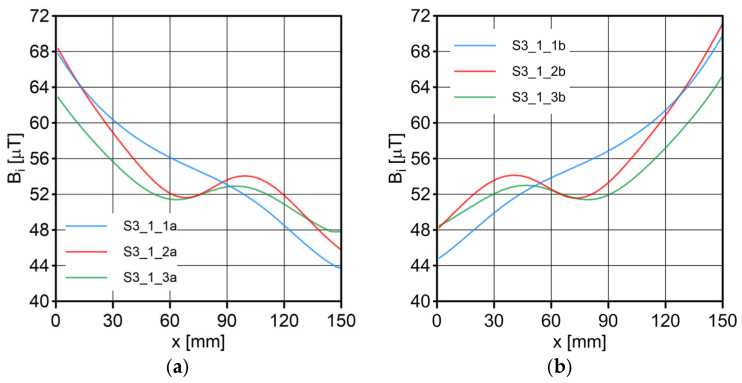
Distribution of the magnetic induction *B* components along the rope—sensor S3_1: (**a**) forward direction; (**b**) backward direction.

**Figure 14 sensors-23-06365-f014:**
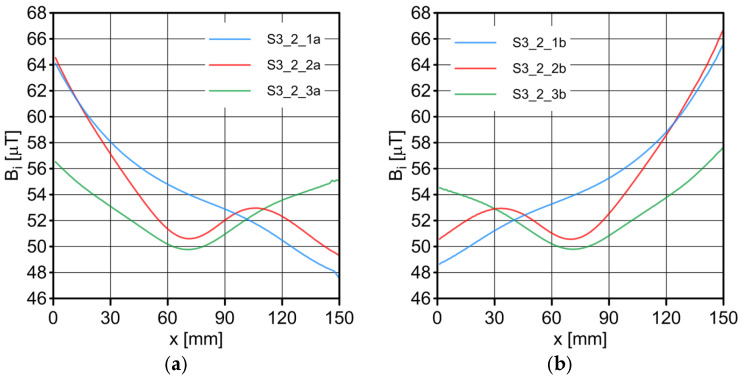
Distribution of the magnetic induction *B* components along the rope—sensor S3_2: (**a**) forward direction; (**b**) backward direction.

**Figure 15 sensors-23-06365-f015:**
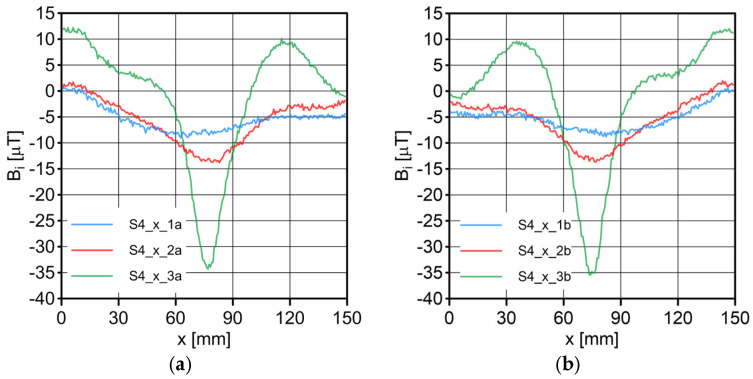
Distribution of the magnetic induction *B* components along the rope—sensor S4, along the x-axis: (**a**) forward direction; (**b**) backward direction.

**Figure 16 sensors-23-06365-f016:**
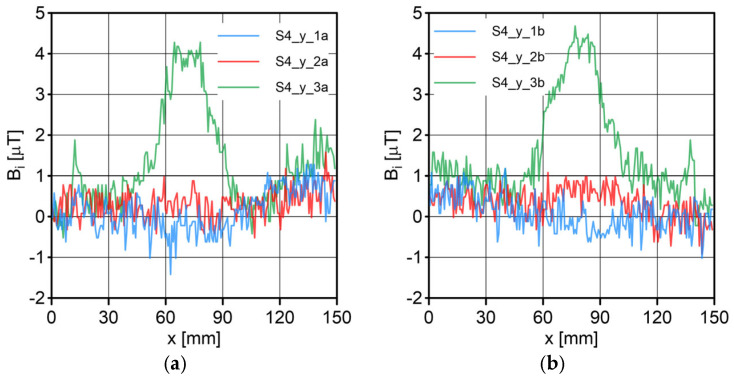
Distribution of the magnetic induction *B* components along the rope—sensor S4, along the y-axis: (**a**) forward direction; (**b**) backward direction.

**Figure 17 sensors-23-06365-f017:**
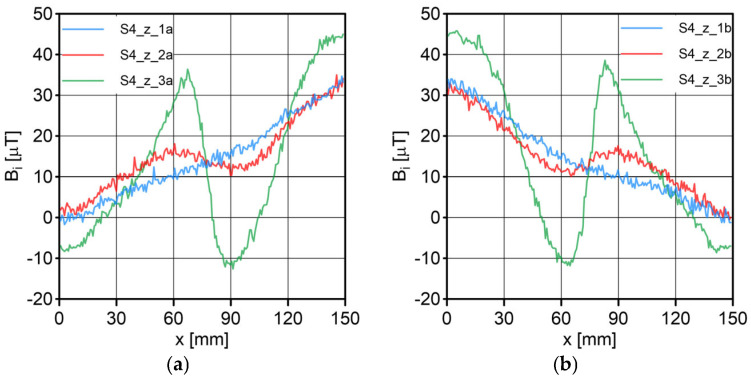
Distribution of the magnetic induction *B* components along the rope—sensor S4, along the z-axis: (**a**) forward direction; (**b**) backward direction.

**Table 1 sensors-23-06365-t001:** Characteristics of the sensors used.

No.	Manufacturer’s Designation	Producer	Measurement Phenomenon Used	Sensor Characteristics (According to Information from Producers)			Comments
				Typical Scale Range	Sensitivity	Noise	
**S1**	MI-CB-1DJ-S-B-USB	Aichi Steel (Japan)	One-axis magnetoimpedance magnetometer	2μT pp	5.0 V/μT	1 nT/1σ	Single sensing element
**S2**	MI-CB-1DJ-D-B-USB	Aichi Steel (Japan)	One-axis magnetoimpedance magnetometer				A line of differential sensing elements
**S3**	Micro-Fabricated Atomic Magnetometer	Geometrics (USA)	Miniature scalar magnetometer evaluation module with two laser-pumped cesium sensors	20–100 µT	1 pT/$\sqrt{\mathrm{Hz}}$	5 pT/$\sqrt{\mathrm{Hz}}$	It consists of two independent sensors labeled S3_1 and S3_2
**S4**	SpinMeter-3D USB 3 Axis Magnetometer	Micro Magnetics (USA)	Digital triaxial magnetometer based on quantum magnetic tunnel junctions (MTJs) and magnetoresistance tunneling effect (TMR)	±1000 µT	0.10 µT	0.25 µT rms (minimum)	Measures the value of magnetic induction in 3 axes marked as S4_x, S4_y, S4_z

## Data Availability

Not applicable.

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
