# Peer review of "Use of Different Types of Magnetic Field Sensors in Diagnosing the State of Ferromagnetic Elements Based on Residual Magnetic Field Measurements"

_sensors, 2023, doi:10.3390/s23146365_

Round 1

Reviewer 1 Report

[1].   For the differences in the acquisition results of different sensors, the authors can use the acquisition principles of different sensors to analyze the main reasons for the different results.

[2].   Different sensor packages are different. The packaging process of the magnetic sensor itself can bring about lift off differences. Do these differences need to be considered in the analysis of the results.

[3].   In this paper, four sensors are used for simultaneous tesing signals. Is there any interference in the signals from different sensors?

[4].   There are still some more important works, such as magnetic memory testing and other magnetic flux leakage testing technologies. It is recommended to add some references in the introduction.

Minor editing of English language required

Reviewer 2 Report

The manuscript deals with the study of the possibility of detecting mechanically introduced discontinuities using different magnetic sensors. Despite the large amount of information presented, this work cannot be published in this form for the following reasons:

1. The abstract does not reflect the work done.

2. The introduction contains a lot of redundant information, moreover, section 2.2. looks like part of the introduction.

3. Section 3.1. Table 1 does not contain the characteristics of the sensors used, namely: sensor sizes, noise level, sensitivity, dynamic range, etc.

4. Section 3.2. Figures 2-6 are not informative, since they contain information about another magnetic field range (> 300 µT), but measured signals are in the range of less than 100 µT.

5. It is not clear how the “measurement range” (150 mm) in Figs 8-10 relates to the X coordinate (150 mm) in Figs 11-14.

6. The authors cannot explain the appearance of the three peaks in Figures 11-12. One of the possible explanations is the influence of other sensors on the measurement result (this needs to be checked).

Reviewer 3 Report

Nondestructive testing of steel ropes is a very important and up to date problem. Magnetic methods seem to be a useful tool for this problem, because of the ferromagnetic steel. There exist several methods in the literature for solving this problem, and there is a wide choice of device in the market for doing this inspection. Passive techniques are becoming increasingly popular, using a residual magnetic field and not requiring magnetization of the tested object with a strong external field. It is possible to find a correlation between the self-magnetization process in ferromagnetic materials and defects steel wire ropes. Authors try to contribute for this problem by using different types of magnetic field sensors in diagnosing the state of ferromagnetic elements based on residual magnetic field measurements. However, in my opinion, the content of novelty of their work is rather low and I do not see how their results could be useful in practical application. Because of this I suggest to reject the manuscript.

Arguments are the following:

Authors give a good and comprehensive summary of research articles, where the problem is discussed and methods are suggested for steel wire rope testing, and call attention for the weakness of these methods. But they do not propose any technique, which can be at least competitive with them. After this careful analysis of weaknesses of existing methods I would expect something novel in the investigation technology, advantageous compared with the refereed methods.

Moreover, apart from the referenced articles, many devices, techniques exist in the market for solving this problem. (Just an example: Kazimierz Zawada: Magnetic NDT of Steel Wire Ropes.) Are all the existing techniques also not suitable for a reliable test of ferromagnetic steel ropes?

The only thing that authors did, that they applied four different magnetic field sensors and they compared the signals of these sensors. It is not explained, why exactly these four sensors have been chosen. Are the sensors the same, which are used in exisiting techiques? Or they are different? Better, than others?

A significant weakness of the manuscript is that the experimental condition, the property of investigated rope is not described. Measurement results are presented on a rope, and something is detected, but the details of experiment is not given. Authors say: „The sensor registered changes in induction in the vicinity of the discontinuity”. But it is not shown, what was the discontinuity. Was it some artifcial crack or someting else? If the defect is not known, it is not possible to say anything about the sensitivity and efficiency of its detection. A big defect can be detected easily by any method.

It is known that SMFL method relies on the self-magnetization of ferromagnetic material in a geomagnetic field, while the MFL method requires an externally formatted excitation source. This means that the SMFL experiment is simple, because there is no need to apply strong magnetic field. But on the other side, it means that the self-magnetization of the investigated object highly depends on the direction of the rope with respect to the North-South direction. This is the weakness of this method. How do the authors take into account the position of the examined rope? What happens if the rope is rotated with respect to the N-S direction? The same question arises if the inspection is made not in the laboratoy but in the field.

Some minor remarks:

It would be nice if the authors explain a little bit more detailed Fig.1. The structure of the steel wire rope is not understandable for not experts in this area.

Authors write: „In series 1 of the measurement, the steel wire rope was installed on the delivery state  on the stand according to Fig. 3.  But Figure 3. shows the Helmholtz coil calibration curves.

The S4 sensor is a low-power, 3-axis USB digital vector triaxial magnetometer based  on magnetoresistance tunnelling effect (TMR) and quantum magnetic tunnel junctions  (MTJ). What are the details of this sensor?

Round 2

Reviewer 1 Report

The authors have made careful revisions to the review comments, and the quality of the paper has been greatly improved. It is recommended to accept it. 

Author Response

We are very grateful for the review effort.

Reviewer 2 Report

The last version of the manuscript has been improved and can be accepted after the minor corrections:
1.     Figure 2 does not affect the content and is redundant in this work.
2.     For convenience of comparison of magnetic data in Fig. 11-15 (for direct and reverse measurements) the corresponding Y-axis should have the same range (for example: -1.2/+1.6 for Fig. 11).
3.    The characteristics of the sensors in Table 1 are incorrect and should be corrected.

Author Response

We are very grateful for the review effort. See attached file.

Reviewer 3 Report

Authors have made improvements in the manuscript, which are acceptable, but one point. I have critized the lack of the description of defect. I wrote in my review: „But it is not shown, what was the discontinuity. Was it some artifcial crack or someting else? If the defect is not known, it is not possible to say anything about the sensitivity and efficiency of its detection. A big defect can be detected easily by any method”.

Authors did not give satisfactory information about the defect in the revised mauscript, as well. They write „Defect introduced in the rope that was artificial discontinuity which size was presented on the Fig. 9-10.” But I do not see in these figures how the artificial defect looks like. Size? Shape? A photo about defect would be helpful. And some words about this type of defect, wether it is typical or not in everyday practice.

Author Response

We are very grateful for the review effort. See atached file.
